# COPSOQ III in China: Preliminary Validation of an International Instrument to Measure Psychosocial Work Factors

**DOI:** 10.3390/healthcare13070825

**Published:** 2025-04-04

**Authors:** Yiming Huang, Min Zhang, Fuyuan Wang, Yuting Tang, Chuning He, Xinxin Fang, Xuechun Wang

**Affiliations:** School of Population Medicine and Public Health, Chinese Academy of Medical Sciences & Peking Union Medical College, Beijing 100730, China; huangyiming@sph.pumc.edu.cn (Y.H.); wangfuyuan@student.pumc.edu.cn (F.W.); tyting0109@student.pumc.edu.cn (Y.T.); hechuning@student.pumc.edu.cn (C.H.); s2023027004@student.pumc.edu.cn (X.F.); wangxuechun@student.pumc.edu.cn (X.W.)

**Keywords:** COPSOQ III, psychosocial work factor, psychometric evaluation, health workers, occupational health

## Abstract

Background: The Copenhagen Psychosocial Questionnaire (COPSOQ) is an international instrument designed for the assessment and improvement of psychosocial conditions in workplaces and for research purposes. After the International COPSOQ Network took over responsibility for the development of COPSOQ, a new version was published in 2019 (COPSOQ III). However, there is no widely recognized instrument for measuring psychosocial work factors, and insufficient research exists regarding the use of COPSOQ III within the Chinese setting. This study seeks to conduct a preliminary validation of the Chinese long version of COPSOQ III. Methods: A cross-sectional questionnaire survey was conducted with a large sample in a medical consortium in China. Item analysis, reliability, and validity were explored. Results: The data management platform showed that 1067 respondents who met the inclusion criteria completed the questionnaire, among which 1054 questionnaires were valid. The content validity results showed that the I-CVI and S-CVI/AVE of each item ranged from 42.86 to 100.00, and the S-CVI/AVE was 92.52, indicating that the Chinese long version of COPSOQ III had good content validity. An exploratory factor analysis (EFA) extracted seven factors, which explained 63.96% of the variation. The results of confirmatory factor analysis (CFA) indicated that the model was well fit (χ^2^/df = 4.50, CFI = 0.910, TLI = 0.897, IFI = 0.910, RMSEA = 0.058). The Cronbach alpha coefficient of the questionnaire was 0.92. Conclusions: The Chinese long version of COPSOQ III is of good reliability and validity, like the original, and can be used to assess psychosocial work factors in China.

## 1. Introduction

Psychosocial work factors have become one of the basic elements of the current and future working environment, and their impact on the health of the occupational population has been of increasing concern. COVID-19 has given rise to or accelerated the development of new technologies, represented by digitalization and communication technologies, and new ways of working, represented by working from home, which has increased workers’ psychosocial problems [1].

The workplace factors that can cause stress are called psychosocial hazards [2,3,4]. In 1984, the International Labour Organization (ILO) defined psychosocial factors (hazards) as the interactions between and among work environments; job content; organizational conditions; and workers’ capacities, needs, cultures, and personal extra-job considerations. They could influence workers’ health, work performance, and job satisfaction through perceptions and experience in the workplace. This definition emphasizes the dynamic interaction between the work environment and human factors. Risks to mental health—also called psychosocial risks—may be related to job content, work schedule, specific characteristics of the workplace, or opportunities for career development, among other things [5]. Exposure to psychosocial hazards can lead to serious physical and psychological complications. These hazards include discrimination, inequality, overwhelming workload, and job insecurity. If the body does not recover adequately, the risks increase for conditions such as cardiovascular diseases [6], burnout [7], and musculoskeletal disorders [8]. Other potential effects include depressive symptoms [9], dementia [10], sleep disturbances [11], post-traumatic stress disorder (PTSD) [12], and even suicide [13]. It is worth pointing out that the ILO List of Occupational Diseases (revised in 2010) included mental and behavioral disorders, including post-traumatic stress disorder [14].

In 2021, the International Commission on Occupational Health (ICOH) estimated that the total incidence of work-related mental health/disorders (stress, night work, and psychosocial factors) in the world accounted for 10% to 30% of such disorders [15], and the total number of diseases has been estimated at 173 million in China [16]. Specifically, the number of work-related mental health/disorders (stress, night work, and psychosocial factors) is about 17.3 million to 51.9 million in China. The impact of work-related stress on workplace productivity and the broader economy is also considerable, causing huge losses and injuries in terms of labor participation, production efficiency, and quality of work [17,18].

The Copenhagen Psychosocial Questionnaire (COPSOQ) is an international instrument designed to assess and improve psychosocial conditions in workplaces and for research purposes. COPSOQ is one of the most widely used psychosocial risk assessments, and it has been cited as a reference in documents of international organizations such as the United Nations (UN) [19], the World Health Organization (WHO) [20], and the ILO [21], and is recognized as an example of good practice by the EU Occupational Safety and Health Agency [22]. COPSOQ was developed around the year 2000 and is available in more than 25 languages for risk assessments of psychosocial factors and scientific research projects, allowing for the improvement of the corresponding intervention evaluations [23]. COPSOQ integrates the most prominent work environment theories, including demand–control–social support, effort–rewards, job demands–resources, work–family conflict, social capital, socio-technical theory, etc. The demand–control–social support model and the effort–reward model are the predominant frameworks for understanding job stress in the workplace. The demand–control–social support model quantifies job stress by integrating job demand elements (workload, physical demands, time pressure, and breaks), job control (autonomy), and social support (from supervisors and colleagues) [24]. The effort–rewards model holds that a disparity between the effort exerted at work and the rewards obtained (such as money, dignity, job opportunity, or job security) leads to emotional distress, heightening the risk of adverse health outcomes [25]. The job demands–resources theory proposes that employees exhibiting work engagement—characterized by elevated energy, dedication, and absorption—actively seek to enhance their job demands and resources through job crafting. In contrast, employees experiencing job strain will begin to undermine their own performance at work. [26]. Work–family conflict arises when the demands of the work and family domains are incompatible, thereby impeding the performance of the domains. Conflict can arise in two ways: either work can impede the ability to meet family demands or family can impede the ability to meet work demands [27,28]. Social capital originated from the domain of sociology and is regarded as an important element for organizational success through networks of relationships [29]. Socio-technical theory is used to analyze intricate organizational structures that emphasize the interaction between human resources and technology/social and technical systems in the workplace [30]. No comprehensive theory or model currently encompasses all significant facets of the psychosocial work environment. COPSOQ synthesizes various existing models to create a more robust, holistic instrument for addressing the hitherto ignored aspects of the worker and interaction with the workplace environment. This instrument was revised in 2010, giving rise to COPSOQ II [31]. The International COPSOQ Network (http://www.copsoq-network.org) has collaboratively organized its development since 2007 according to the principles of action-oriented research [32]. The international network is responsible for regular updating and adaptation to labor market changes and scientific progress. It is also responsible for reaching a consensus on the use of COPSOQ III regarding the definitions, dimensions, items, and criteria to guarantee international and longitudinal comparability. In each country, the Network recognizes a “national COPSOQ team”, i.e., the team that adapts and validates COPSOQ to the country and language. When applying it to any country, what is required is the uniformity of the national short, middle, and long versions and clarity of language expression. The long version is mainly intended for research purposes and for providing opportunities for national adaptations depending on what is needed in the context, with the medium-length version to be used by work environment professionals and the short version for workplaces. The core items are mandatory for all national versions of any length to allow for international and longitudinal comparisons. All of the questionnaire items have a Likert-like scale and values for the coding of responses. Except for the question on general health, which uses a ten-interval answer scale, there are always five possible answers, with values corresponding to 0, 25, 50, 75, and 100 points.

Research on psychosocial work factors in China commenced in the early 1990s, concentrating on occupational stress and its effects on health [33]. There is a dearth of studies on psychological work factors, including risk assessment, intervention strategies, and the formulation of regulations and standards. In 2005, Li Jian introduced COPSOQ I to China for the first time and provided a new instrument for psychosocial work factors. The short version of the Chinese COPSOQ I has been validated with good reliability and validity [34]. It has been used for measuring psychosocial factors in various working populations after revising some items. A study of gender differences under the effect of the psychosocial work environment on health functioning in Chinese occupational populations showed that psychosocial work factors had negative effects on health functioning and that women were more vulnerable to work stress than men [35]. Additionally, the Chinese Nurses’ Early Exit Study (NEXT Study) used COPSOQ I to investigate the relationship between poor psychosocial work conditions and intention to leave [36], burnout [37], needle stick and sharps injuries (NSIs) [38], health functioning [39], occupational stress [40], and suboptimal health status [41] among the occupational population. Subsequently, the team also sinicized and introduced COPSOQ II.

Occupational health for HWs in China has been particularly emphasized since the SARS outbreak in 2003 [42]. The progress of occupational health for HWs can be divided into three stages. The first stage (2003–2008) focused on policy analysis and the development of evidence-based national guidelines: the Guideline for Prevention and Control of Occupational Exposure to Bloodborne Pathogens. The second stage (2009–2012) focused on the implementation of the guidelines and hospital intervention pilots. The third stage (2013–2019) concentrated on a methodical approach to improving HWs’ health and well-being. By the end of 2019, the occupational health of HWs in China was integrated into a thematic focus on measures from bloodborne pathogens to workplace violence toward HWs. Furthermore, risk assessment of psychosocial hazards is rare from the perspective of the Model of Hospital Initiative on Systematic Occupational Health (HISOH Model).

To date, there is no updated COPSOQ III as a widely recognized instrument for measuring psychosocial work factors in China. In March 2023, our research team received authorization from the COPSOQ Network and scholar Li Jian for the validation of COPSOQ III. Building on the success of validating COPSOQ I and COPSOQ II in China, this study aims to present, discuss, and evaluate the Chinese long-version of COPSOQ III in terms of its cross-cultural adaptation, reliability, and preliminary validity among HWs in a Chinese medical consortium.

## 2. Materials and Methods

### 2.1. Study Design

This is an exploratory cross-sectional study aiming at the cross-cultural adaption of COPSOQ III to China by employing item analysis and tests of reliability, content validity, and structural validity to reduce the number of items and to obtain insight into the data structure. These phases follow the international procedures defined by the International COPSOQ network.

### 2.2. Study Setting and Participants

In January 2013, the National Health Commission first mentioned the concept of the medical consortium at the government level [43]. The medical consortium refers to tertiary hospitals as the core, combined with secondary hospitals in the region and primary hospitals. It can be divided into urban medical groups, county medical communities, telemedicine collaboration networks, and specialist alliances [44].

The core tertiary hospital (Hospital A) specializes in the treatment of infectious diseases and comorbidities, developing the HISOH Model [45]. According to the principle of stratified sampling and the recommendation of the core tertiary hospital, we selected a secondary hospital (Hospital B) that specializes in the treatment of psychiatric and infectious diseases and two primary hospitals (Hospitals C and D) focusing on basic public health services and medical care.

Considering the sample medical consortium as a cluster, we involved all health workers (HWs) who met the inclusion criteria as participants. The inclusion criteria of participants were as follows: (1) HW with professional certification, (2) voluntary participation in the survey with informed consent, and (3) employed by the hospital as a regular employee for >1 year. Exclusion criteria were as follows: (1) those failing to answer the questionnaire in the opening hours and (2) those exceeding the time limit for the questionnaire.

### 2.3. The Questionnaire Content of the Chinese Long Version of COPSOQ III

The Chinese long version of COPSOQ III includes two parts: (1) socio-demographic characteristics: age, gender, ethnic group, marital status, educational attainment, position, occupation, etc.; and (2) psychosocial work factors: quantitative demands, work pace, cognitive demands, emotional demands, demands for hiding emotions, influence at work, possibilities for development, variation of work, control over working time, meaning of work, predictability, recognition, role clarity, role conflicts, illegitimate tasks, quality of leadership, social support from supervisor, social support from colleagues, sense of community at work, commitment to the workplace, work engagement, job insecurity, insecurity over working conditions, quality of work, job satisfaction, work—life conflict, horizontal trust, vertical trust, organizational justice, gossip and slander, conflicts and quarrels, unpleasant teasing, cyber bullying, sexual harassment, threats of violence, physical violence, bullying, self-rated health, sleeping troubles, burnout, stress, somatic stress, cognitive stress, depressive symptoms, and self-efficacy.

Except for the question on general health, which uses a ten-interval answer scale, all of the questionnaire items have a 5-points Likert scale from “never (0)” to “always (100)” and from “none (0)” to “extremely (100)”. The core items in the short, medium, and long versions of COPSOQ III are mandatory and cannot be removed.

### 2.4. Translation

Adhering to the Brislin model, the translation was executed through a systematic approach comprising sequential translation, back-translation, cultural adaptation, and semantic analysis to guarantee the translation’s fidelity to the original text [46,47]. The first translation from English to Chinese was independently conducted by two researchers (Y.H. and C.H.), and the process was guided by the principal investigator (Prof. M.Z.). After consensus was reached on all items, they were sent to another two researchers (F.W. and Y.T.) to be back-translated into English. The English back-translation was then compared with the original version, and revisions were made to the final version. Finally, to avoid potential linguistic problems in the Chinese version, an experienced professional linguist thoroughly reviewed the questionnaire. The Chinese long version of COPSOQ III underwent cross-cultural adaptation [48] through a workshop. All participants from the research team involved in translation are bilingual and experts in public health. Items with varying interpretations were debated and changed based on country circumstances and cultural variances, all while maintaining the original questionnaire concept. We selected ten health workers from a hospital for the semantic analysis. After making modifications based on feedback, including avoiding all dialect-specific words and maintaining an informal oral style, the Chinese long version of COPSOQ III was completed.

### 2.5. Data Collection

The researchers obtained permission from the participants in the workplaces. Members of the team visited the relevant departments and continued to invite on-duty HWs to fill out an online questionnaire using their cell phones in July 2023. The questionnaire is closed, and all items are designated as mandatory. Moreover, the participants were informed that the study would be conducted voluntarily, and their consent was received. The informed consent was presented on the first page of the questionnaire, which could only be accessed by those who had given their informed consent. The Chinese long version of COPSOQ III is available online (https://www.wjx.cn/vm/YGWJ4HB.aspx, accessed on 24 December 2024).

### 2.6. Statistical Analyses

Descriptive statistics were calculated for the demographic and frequency of participants. Distributional descriptive statistics of dimension characteristics, including mean, standard deviation, and floor and ceiling distribution, were calculated. Floor and ceiling effects are considered to be present if more than 15% of respondents achieved the lowest or highest possible score, respectively [49]. If floor or ceiling effects are present, extreme items are likely missing at the lower or upper end of the scale, indicating limited validity of the content [50].

The extreme group comparison method was used for item analysis of the questionnaire. An independent-sample T-test was used to compare the differences in all items between the high group (top 27% of subjects, *n* = 285) and the low group (bottom 27% of subjects, *n* = 285). If the results show that there is no significant difference in the average score of each item between the two groups with high and low scores, the item should be deleted [51]. The CR value reaching the significance threshold with a value greater than 3 indicates that the measured items have parameter discrimination [52]. Internal consistency and reliability were analyzed with Cronbach’s alpha coefficient for scales with three or more items and the Spearman–Brown coefficient for two-item scales [53]. It is generally believed that when Cronbach’s alpha coefficient ≥0.70, the reliability is acceptable, while items with a Cronbach’s alpha coefficient <0.50 should be deleted [54]. During the content validity evaluation process, the selection criteria of experts participants in this study were a bachelor’s degree or above, senior professional title, and multidisciplinary knowledge of occupational health protection for HWs. Each expert used a 5-point Likert scale ranging from “very irrelevant (1)” to “very relevant (4)”. The Content Validity Index (CVI) includes the Item-level CVI (I-CVI) and Scale-level CVI averaging calculation (S-CVI/Ave). It is generally believed that I-CVI is 0.78 and S-CVI/Ave is 0.90, indicating that the content validity of the questionnaire is good [55].

To determine the construct validity of the questionnaire, the study carried out exploratory factor analysis (EFA). EFA was performed using principal component analysis (PCA) for extraction and Varimax rotation with Kaiser normalization to achieve a clear and interpretable factor structure. The Kaiser–Meyer–Olkin (KMO) coefficient and the Barlett test were used to determine whether or not the data were fit for factor analysis prior to the analyses. A value of KMO > 0.8 and a statistically significant result on Bartlett’s sphericity test indicate that scales are suitable for factor analysis [56]. Factor extraction was carried out with and eigenvalue ≥ 1 [57]. The criteria for item elimination were based on multiple considerations: (1) items with factor loadings below 0.40 were removed to ensure strong associations with the intended factors, (2) items with cross-loadings exceeding 0.30 on multiple factors were excluded to maintain discriminant validity, and (3) items that did not contribute meaningfully to the theoretical interpretation of the factors were discarded [58].

The model compliance of COPSOQ III was assessed using first-level confirmatory factor analysis (CFA). A CFA using a weighted least-square mean- and variance-adjusted estimator (WLSMV) was also conducted. The overall goodness of fit was assessed using the following indices and cut-off points for “good adjustment”: chi-square (χ2), comparative fit index (CFI; 0.90 ≤ CFI ≤ 0.95), and Tucker–Lewis index (TLI; 0.90 ≤ TLI ≤ 0.95). For RMSEA, a value of <0.08 was the acceptable limit, and <0.50 was the excellent-fit limit [59,60]. Convergent validity refers to how well sub-scales correlate with other measures that are assumed to be related. This criterion assesses the convergent validity of the measurement model through the Average Variance Extracted (AVE) and Composite Reliability (CR). AVE measures the level of variance captured by a construct versus the level due to measurement error; values above 0.7 are considered excellent, whereas a level of 0.5 is acceptable. CR is a less biased estimate of reliability than Cronbach’s alpha coefficient; the acceptable value of CR is 0.7 and above [61,62]. Discriminant validity evaluates the ability to discriminate between groups with known differences. The square root of the AVE for the variables was greater than the correlation between the dimensions in the model, supporting the scale’s discriminant validity. All analyses were conducted using IBM SPSS Statistics V.26.0 (SPSS Inc., Chicago, IL, USA) and IBM SPSS Amos version 26.0 (Chicago, IL, USA).

## 3. Results

### 3.1. The Chinese Long Version of COPSOQ III

After sequential translation, back-translation, cultural adaptation, and semantic analysis, the long version of COPSOQ III was translated from English into Chinese. We revised expressions that deviate from Chinese linguistic conventions to uphold the original meaning of the scale, enhancing the translation’s accuracy and professionalism. As shown in Appendix A, the original questionnaire, with all entries preserved, formed the Chinese long-version of COPSOQ III.

### 3.2. Socio-Demographic Characteristics of the Sample

The number of HWs who met the inclusion criteria was 1160. The data management platform showed that 1067 respondents who met the inclusion criteria completed the questionnaire, of whom 1054 had valid questionnaires (total response rate: 91.98%; total valid response rate: 98.78%). As shown in Table 1, the mean age and standard deviation of the overall sample were 33.77 ± 9.27. Furthermore, 74.00% of the participants were female, and 26.00% were male. This study also found that 20.59% of the participants worked in a Grade 1A hospital, 19.07% in a Grade 2A hospital, and 60.25% in a Grade 3A hospital. Of the participants, 23.91% were doctors, 47.44% were nurses, and 28.65% were technical support and administration staff.

### 3.3. Descriptive Statistics of Scales

For each scale, the mean; standard deviation; and fractions with ceiling, floor, and missing values were calculated to check for sensitivity and variation. As shown in Table 2, the mean values of the Chinese long version of COPSOQ III varied from 26.67 points for “somatic stress” to 72.02 points for “meaning of work”. The standard deviations of all scales ranged from a minimum of 16.25 points to a maximum of 23.28 points. These values cannot be interpreted in a normative way. But COPSOQ guidelines are not fixated upon any cut-off values, however legitimate, as “the true” values. Floor effects—defined here as the percentage of answers coded zero—ranged between 0.09 and 15.56%. There was one scale with 20% and more in this category (“meaning of work”), while 23 scales had less than 5% answers at this extreme. Ceiling effects—defined as the percentage of answers coded 100—ranged between 0.09 and 15.56%.

The scores of male HWs were significantly higher than those of female HWs in the following dimensions: quantitative demands, emotional demands, influence at work, control of working time, role conflict, illegitimate tasks, work–life conflict, cognitive stress, and depressive symptoms (*p* < 0.05). The scores of female HWs were statistically significantly higher than those of male HWs in the following dimensions: development possibility, meaning of work, role clarity, social support from colleagues, sense of community at work, commitment to the workplace, job satisfaction, and organizational justice (*p* < 0.05). These results prove significant differences in perception of males and females under various conditions of work and society.

### 3.4. Item Analysis

As shown in Appendix A, the independent-samples T test shows that there was no significant difference in means between the two groups with high and low scores (*p* < 0.001), and the CR value of each item was much higher than three, which indicates that the Chinese long version of COPSOQ III has a good discrimination ability.

### 3.5. The Results of Validity and Reliability

This section delineates the two primary phases of the study: content validation and the trimming process, which facilitated the assessment of reliability and validity, respectively, with the objective of item elimination and the attainment of a research version.

#### 3.5.1. Content Validity

The evaluation of content validity involved seven experts occupying senior professional positions. Six of the experts were female, with ages ranging from 44 to 60 years. The discipline and research area of each expert are shown in Table 3.

As shown in Appendix A, the content validity evaluation results show that the I-CVI and S-CVI/Ave of the Chinese long version of COPSOQ range from 42.86 to 100.00, and the S-CVI/AVE is 92.52, indicating that the questionnaire has good content validity. For 16 items whose I-CVI is less than 0.78, considering that the core items (MW1 and CO2) on the original scale cannot be deleted, only 14 items are deleted, namely IN2, IN3, IN4, IN5, IN6, CT4, MW2, QLX1, SC3, SW3, JS5, WFX1, CS1, and DS2.

#### 3.5.2. Construct Validity

EFA is an appropriate means to check statistical relations for a multitude of scales. The results of the first phase of EFA showed that the KMO coefficient = 0.93 and the Bartlett sphericity test χ^2^ value was 22,033.29 (*p* < 0.001), indicating that the variables were not independent. Seven components were identified, accounting for 63.65% of the total variation. Obviously, there is a certain fuzziness between components such as “influence at work” and “possibility of development” loading on both components, while “self-rated health” proved to be an invalid item. The following items were eliminated after retaining the essential items of the original questionnaire: IN2–IN6, PD4, TE1–TE3, TM1–TM4, GH2, WE1–WE3, and SE1–SE4. In the second phase of EFA, the KMO coefficient was 0.92, and the Bartlett test result was *p* < 0.001 (χ^2^ = 18,179.18, *df* = 528). Because the KMO value was higher than 0.60 and the result of Bartlett test was significant, the data were suitable for factor analysis. Seven components were also extracted, which explained 63.96% of the total variation.

As shown in Table 4, the factors numbered 1–3 combined a larger number of scales than factors 4–7. Component 1 showed high loadings for “recognition”, “role clarity”, and “commitment to the workplace” and could therefore be called “interpersonal relations and leadership”. Component 2 strongly connected “cognitive stress”, “stress”, “depressive symptoms”, “somatic stress”, “burnout”, and “sleeping troubles” and could represent the domain of “health and well-being”. Component 3 combined “work pace”, “cognitive demands”, “emotional demands”, “demands for hiding emotions”, and “quantitative demands” into “demands at work”. Component 4 represented “work organization and job contents”, combining “control over working time” and “variation of work”. Component 5 represented the “negative tasks”, combining “role conflicts” and “illegitimate tasks”. Component 6 represented “workplace insecurity”, combining “job insecurity” and “insecurity over working conditions”. Component 7 represented “developmental influence”, combining “possibilities for development” and “influence at work”. The results of CFA also indicated that the model exhibited goodness of fit (χ^2^/df = 4.50, CFI = 0.91, TLI = 0.90, IFI = 0.91, RMSEA = 0.06). It is possible to verify that both the EFA and CFA supported the decisions made in the trimming process.

#### 3.5.3. Convergent and Discriminant Validity

As shown in Table 5, the CR values of the seven factors ranged from 0.76 to 0.97, and the AVE values ranged from 0.52 to 0.85, indicating that the Chinese long version of COPSOQ III had good convergent validity.

The results of the discriminant validity analysis are presented in Table 6. The square-root values of AVE ranged from 0.53 to 0.79. For all seven factors, the AVE square-root values were greater than the maximum absolute value of the correlation coefficients between factors. These results indicate that the Chinese long version of COPSOQ III demonstrated good differentiation effectiveness.

#### 3.5.4. Reliability Analysis

As shown in Appendix A, the Chinese long version of COPSOQ III demonstrated satisfactory reliability, as evidenced by a Cronbach Alpha coefficient of 0.92 for the overall questionnaire and 0.60 to 0.92 for each dimension. Furthermore, the analysis of Cronbach’s Alpha coefficient upon item deletion for QD4, HE4, CT5, CW5, IW5, WFX1, TE1, and TM4 indicated that the dimension corresponding to each item exhibited a higher Cronbach’s Alpha coefficient value.

## 4. Discussion

COPSOQ III, as an instrument for assessing and preventing psychosocial work risks and fostering organizational development, requires adaptation to the specific contextual conditions of different countries and must be validated. The COPSOQ instrument covers a broad range of domains, including demands at work, work organization and job contents, interpersonal relations and leadership, work-individual interface, social capital, offensive behaviors, and health and well-being. In the context of the absence of international instruments to assess psychosocial work factors in China and considering the convenience of sample acquisition, this study conducted psychometric testing of COPSOQ III in a Chinese medical consortium for the first time.

### 4.1. Upgrading the First Chinese Long Version of COPSOQ III

The participants were heterogeneous in terms of several characteristics, including age, educational attainment, contract status, and occupational classification. Floor effects—defined here as the percentage of answers coded zero—ranged between 0.0 and 23.72%. Ceiling effects—defined as the percentage of answers coded 100—ranged between 0.09 and 25.80%. This study found that the floor and ceiling distributions of the dimensions were distant from the extreme values, except for “meaning of work”, “somatic stress”, and “depression”. These results are acceptable, since there is room for variation in dimension scores.

Since there are no accepted standards (not fixated upon any cut-off values) for workers’ exposure to psychosocial risk factors, comparing the COPSOQ III results to general population reference values is an appropriate and recognized way to classify the significance of such exposures. The mean values of the Chinese long version of COPSOQ III varied from 26.67 points for “somatic stress” to 72.02 points for “meaning of work”. The standard deviations ranged from a minimum of 14.84 points to a maximum of 23.28 points. In comparison, the results of an international survey conducted in Canada, Spain, France, Germany, Sweden, and Turkey in 2016–2017 [63] indicated mean values ranging within an interval of 12.00 (job insecurity) – 82.00 (social support from supervisor) points. Moreover, the validation results for the German COPSOQ III obtained in March 2020 indicated mean values of scales ranging within an interval of 20.00 (intention to leave profession/job)~77.00 (sense of community) points. With standard deviations, these values ranged within an interval of 16.90–28.40 points [64]. In summary, sensitivity, variance, and distribution characteristics seemed to be of good quality in COPSOQ III in this study.

### 4.2. The Empirical Validity and Reliability Results of Chinese COPSOQ III

Three key concerns drove the implementation of a trimming procedure to make sure the questionnaire was both culturally appropriate and psychometrically sound enough for the Chinese context. First, cultural differences played a significant role, as some items were deemed culturally inappropriate or difficult to comprehend within the Chinese context, potentially leading to inaccurate responses from participants. Second, expert opinions obtained through panel reviews and focus-group discussions indicated that some items were considered irrelevant or less significant in the Chinese workplace environment. Third, statistical outcomes of pilot testing revealed that certain items exhibited low reliability or validity or demonstrated high correlations with other items, resulting in redundancy.

Initially, we evaluated the content validity of the questionnaire in the validity assessment process. Content validity evaluates how well an instrument covers all relevant parts of the construct it aims to measure [65]. The Content Validity Index (CVI) is a crucial element in the adaptation and validation of measurement instruments, particularly for international research with populations from diverse linguistic backgrounds [66]. The purpose of establishing content validity is to confirm that the Chinese version of the questionnaire adequately captures the intended constructs and is culturally and contextually appropriate for the Chinese linguistic and work environment. Scoring each item of the questionnaire by seven experts resulted in the deletion of 14 items with I-CVI values of less than 0.78 among 10 dimensions: influence at work, control over working time, meaning of work, quality of leadership, social support from colleagues, sense of community at work, job satisfaction, work–life conflict, cognitive stress, and depressive symptoms. Mostly, they were part of already existing dimensions. It should be noted that since the core items are considered mandatory in the international COPSOQ network to ensure comparability between countries, the “core items” in the above-deleted dimensions were retained. Similar issues occur in established versions from different countries, with practically identical dimensions removed. For instance, COPSOQ III validation studies conducted in Portugal [67] and Turkey [68] excluded the dimensions of work–life conflict, job satisfaction, and control over working time, with the exception of the core items. Furthermore, to ensure the content validity of the Chinese version of the questionnaire, we carefully modified the formulation of some items under the guidance of experts to align them with the Chinese cultural and workplace context. This process not only enhanced the cultural relevance of the questionnaire but also ensured that the instrument effectively captures the psychosocial work factors it is designed to measure in the Chinese context.

The study performed EFA to determine the construct validity of the Chinese long version of COPSOQ III, which found 33 dimensions and 89 items that explained 63.96% of the total variance among seven factors. International validation studies have already recommended that the selection of items within the scales be reconsidered, aiming to overcome problems with psychometric robustness [63]. The four dimensions of job engagement, horizontal trust, vertical trust, and self-rated health were deleted, with factor loadings <0.04 and multiple loads. It is important to stress that the results of this exploratory factor analysis are structurally similar to the transcendental dimensions of the international version of COPSOQ III [63]. The Chinese COPSOQ III demonstrated strong convergent and discriminant validity, with CR values exceeding 0.70 and AVE values greater than 0.50 across all factors. These results are consistent with previous validation studies involving municipalities and healthcare settings in Portugal. The data indicate that the Chinese COPSOQ III can capture different but interconnected aspects of psychosocial work factors among HWs in China, making it a reliable tool for this population.

Alpha estimates were all above the usual 0.7 threshold suggested by methodological sources [69]. Field further elaborated on the role of Cronbach’ s Alpha coefficient if Item Deleted (CAID). CAID represents the value of Cronbach’ s Alpha coefficient if a specific item is removed from the scale. If the CAID is greater than current Cronbach’s Alpha coefficient, removing the item would improve the scale’s reliability, suggesting that the item may be inconsistent or redundant and should be removed. In the revised version, eight items (QD4 for “quantitative demands”, HE4 for “demands for hiding emotions”, CT5 for “control over working time”, CW4 for “commitment to the workplace”, IW5 for “insecurity over working conditions”, WFX1 for “work–life conflict”, TE1 for “horizontal trust”, and TM4 for ”vertical trust”) were deleted. In 2005, when the Chinese short version of COPSOQ I was initially introduced, researchers constructed five dimensions with, Cronbach’s α coefficient ranging from 0.48 to 0.84 [34]. Regarding internal consistency, this study revealed that the overall Cronbach’s Alpha coefficient was 0.92, in general, reaching the values recommended in the literature (above 0.70), with the exception of the “offensive behaviors” dimension. This study demonstrates that the Chinese long version of COPSOQ III possesses strong reliability, surpassing that of the Chinese short version of COPSOQ I.

The Comparative Fit Index (CFI) and Tucker–Lewis Index (TLI) values were 0.91 and 0.90, respectively, after the removal of items from the dataset. These values suggest that the latent variables are well represented by their respective indicators and that the model is well-fit to the data. The improved CFA model represents a robust statistical and theoretical framework that accurately represents the constructs, in close alignment with the underlying theoretical framework [63].

### 4.3. Psychometric Testing of COPSOQ III in Diverse Populations and Countries

The empirical results obtained in this study are strongly aligned with the theoretical foundations that underlie the COPSOQ III. The consistent patterns of item loadings and factor structures, which reflect the fundamental dimensions of the psychosocial work environment as originally conceptualized by Kristensen, are indicative of this alignment. For instance, the Chinese COPSOQ III closely mimics the theoretical constructs proposed in the original COPSOQ III framework, with factors such as “demands at work” “work organization and job contents”, and “social capital”. This consistency not only substantiates the theoretical robustness of the COPSOQ III but also emphasizes its cross-cultural applicability, as evidenced by its successful adaptation to the Chinese context.

In recent years, the structural validity of versions of COPSOQ III of different lengths has been investigated in diverse populations in terms of occupation, sector, and country. For example, a medium version of COPSOQ III-TR was validated among four workplaces (call center, hospital, and plastic and metal industries) in Turkey with 25 dimensions and 88 items. Using a corresponding approach, eight factors with five overarching domains were identified in the German standard version of COPSOQ III. The medium version of the Portuguese COPSOQ III with 85 items proved to be a valid preliminary version after the trimming process. An initial validation study of the Norwegian version of the COPSOQ III used a national sample of registered nurses with 20 work environment dimensions and 72 items [70]. A validation study of the enterprise version (2018) of COPSOQ-III was conducted with Spanish professional drivers, 25 dimensions, and 74 items [71]. Additionally, the reliability of the Chinese COPSOQ III in capturing the intended constructs is further confirmed by the high internal consistency of the majority of dimensions. These findings are also in agreement with previous adaptations of COPSOQ II and COPSOQ III in other countries, including the Canadian [72], Brazilian [73], Swedish [74], and Australian [75] versions, which also reported strong theoretical and empirical coherence.

Our team previously published a review that offers an introductory overview of workplace psychosocial factors and recommendations for the assessment, monitoring, and conduct of future scientific research on workplace psychosocial hazards in China [76]. The results in this study further show that there might be slightly different versions of COPSOQ III depending on the process of adaptation and validation at a national level, or towards specific populations. Following the conceptual spirit of the international network, COPSOQ was developed for common and universal use. Nevertheless, COPSOQ consistently allows for data comparison via a common and validated methodological approach. It is a future task to compare the results at the core-item level and to find out whether differences in item values are due to methodological, cultural, or other reasons.

### 4.4. Strengths and Limitations

The current study has several strengths. Firstly, the study applied the questionnaire to participants from health sectors, which promotes the establishment of the HISOH Model and further promotes the prevention and control of psychosocial risk among health workers in China. Secondly, the translation was performed according the Brislin translation model, guided by a multi-disciplinary team of researchers. Our study also has several limitations. Extrapolation to a more extensive population may be inappropriate due to the mere focus on health professionals in a specific medical consortium. However, we believe that this will not have a major impact on the validation results. Furthermore, this study aimed to perform a preliminary assessment of the validity of the scales; therefore, measures such as test–retest reliability, convergent validity, and discriminant validity were not assessed. Validation is an ongoing process, and future validation studies using other occupational settings are needed to improve generalizability, in addition to the need for further analysis of the overall structure of the Chinese long version of COPSOQ III.

## 5. Conclusions

This established Chinese long version of COPSOQ III proved to be the first valid preliminary version for improving the work organization of health workers in the Chinese context. These results offer a foundation for future psychosocial risk assessments and research within Chinese health settings and provide a foundation for the future development of COPSOQ-III in China. Validation is an ongoing process. Future validation studies using other occupational settings are needed to improve generalizability. Moreover, sector-specific measures and benchmarks could be developed for high-risk industries such as services, IT, and platform labor.

## Figures and Tables

**Table 1 healthcare-13-00825-t001:** Socio-demographic statistics for samples (N = 1054).

Variable	Number	%
Gender		
Female	780	74.00
Male	274	26.00
Age (y)		
<35	609	57.78
35~	302	28.65
45~	143	13.57
Ethnic groups		
Han	445	42.22
Zhuang	570	54.08
Others	39	3.70
Marital status		
Single	398	37.76
Married	629	59.68
Others	27	2.56
Types of hospital		
Primary hospital	217	20.59
Secondary hospital	201	19.07
Tertiary hospital	635	60.25
Educational attainment		
Junior college or/and below	322	30.55
Bachelor’s degree	676	64.14
Master’s degree or/and above	56	5.31
Occupation		
Technical support and administration	302	28.65
Doctor	252	23.91
Nurse	500	47.44
Position		
Manager	138	13.09
Employee	916	86.91
Professional title		
Junior or/and below	549	52.09
Middle	340	32.26
Senior	165	15.65
Work tenure (y)		
<10	146	13.85
10~	489	46.39
20~	419	39.75
Contract status		
Permanent	449	42.60
Temporary	605	57.40
Income level (CNY/M)		
<2000	107	10.15
2000~	321	30.46
4000~	322	30.55
6000~	304	28.84
Department		
Technical support and Administration department	382	36.24
Outpatient and emergency department	285	27.04
Ward and other	387	36.72
Commuting time (minutes)		
<30	436	41.37
30~	266	25.24
60~	352	33.39
Shift work		
no	293	27.80
yes	761	72.20
Night work		
no	434	41.18
yes	620	58.82
Direct contact with patients		
no	208	19.73
yes	846	80.27
Intention to leave		
no	938	88.99
yes	116	11.01

**Table 2 healthcare-13-00825-t002:** Descriptive statistics of the Chinese long version of COPSOQ III.

Dimension	Abbr.	Positive Value	No. Of Items	Mean	SD	Ceiling Effect in %	Floor Effect in %
Quantitative Demands	QD	low	4	37.17	17.72	0.09	2.56
Work Pace	WP	low	3	58.86	20.53	4.27	1.42
Cognitive Demands	CD	low	4	59.96	16.84	1.42	0.76
Emotional Demands	ED	low	3	43.52	19.11	0.95	2.75
Demands for Hiding Emotions	HE	low	4	54.49	20.21	1.80	1.23
Influence at Work	IN	high	6	44.68	16.25	0.57	1.23
Possibilities for Development	PD	high	3	62.27	19.20	6.93	0.66
Variation of Work	VA	high	2	40.05	15.74	0.09	2.56
Control over Working time	CT	high	5	38.73	14.84	0.09	0.19
Meaning of Work	MW	high	2	72.02	21.82	23.72	0.95
Predictability	PR	high	2	57.70	21.04	7.40	1.80
Recognition	RE	high	3	62.92	20.96	8.63	1.04
Role Clarity	CL	high	3	69.34	19.65	13.09	0.47
Role Conflicts	CO	low	2	47.72	19.36	2.37	3.80
Illegitimate Tasks	IT	low	1	45.87	23.28	4.74	9.01
Quality of Leadership	QL	high	4	58.70	19.40	5.31	1.04
Social Support from Supervisor	SS	high	3	58.68	20.48	6.55	1.14
Social Support from Colleagues	SC	high	3	62.03	19.45	8.54	0.66
Sense of Community at Work	SW	high	3	68.71	19.18	11.95	0.19
Commitment to the Workplace	CW	high	5	62.88	17.34	3.61	0.09
Work Engagement	WE	high	3	61.01	19.82	4.93	1.71
Job Insecurity	JI	low	3	44.45	21.58	3.13	5.22
Insecurity over Working Conditions	IW	low	5	50.42	18.47	0.66	1.99
Quality of Work	QW	high	2	63.89	17.89	7.12	0.47
Job Satisfaction	JS	high	5	61.76	17.08	3.98	0.47
Work–Life Conflict	WF	low	5	44.28	19.51	1.42	3.32
Horizontal Trust	TE	high	3	60.77	17.59	5.03	0.09
Vertical Trust	TM	high	4	61.39	16.40	3.13	0.09
Organizational Justice	JU	high	4	61.46	19.17	6.07	0.95
Sleeping Troubles	SL	low	4	38.15	22.00	1.04	7.02
Burnout	BO	low	4	37.14	20.84	0.95	6.55
Stress	ST	low	3	36.33	20.17	1.04	6.36
Somatic Stress	SO	low	4	26.67	20.50	0.38	15.56
Cognitive Stress	CS	low	4	31.55	19.84	0.85	9.87
Depressive Symptoms	DS	low	4	28.59	20.48	0.76	15.56
Self-Efficacy	SE	high	6	48.12	20.82	3.61	1.14

**Table 3 healthcare-13-00825-t003:** The discipline and research areas of seven experts.

Expert	Discipline	Research Areas
1	Occupational Medicine	Occupational health policies, regulations, and standards
2	Clinical Medicine	General practice
3	Clinical Medicine	General practice and psychosomatic medicine
4	Nursing	Nursing management
5	Epidemiology	Environmental epidemiology
6	Epidemiology	Human resources for health management
7	Clinical Medicine	Occupational protection for HWs, obstetrics, and gynecology

**Table 4 healthcare-13-00825-t004:** EFA on psychosocial work factors: rotated factor matrix.

Psychosocial Work Factor ^a^	Abbr.	Factor Loadings						
		1	2	3	4	5	6	7
Recognition	RE	0.80	−0.10	−0.05	0.14	−0.02	−0.11	0.27
Role Clarity	CL	0.77	−0.16	0.04	−0.11	0.10	−0.05	0.44
Commitment to the Workplace	CW	0.77	−0.18	−0.06	−0.03	−0.09	−0.11	0.18
Social Support from Supervisor	SS	0.75	−0.02	−0.02	0.19	0.02	−0.08	−0.16
Quality of Leadership	QL	0.75	−0.07	0.06	0.18	0.14	−0.18	0.00
Social Support from Colleagues	SC	0.74	−0.06	−0.02	−0.05	0.07	0.02	−0.18
Sense of Community at Work	SW	0.72	−0.20	−0.04	−0.26	−0.02	0.06	0.02
Job Satisfaction	JS	0.7	−0.16	−0.09	0.15	−0.11	−0.09	0.07
Meaning of Work	MW	0.69	−0.12	0.11	−0.15	−0.01	0.02	0.56
Organizational Justice	JU	0.69	−0.13	−0.01	0.07	−0.23	−0.03	0.02
Quality of Work	QW	0.67	−0.10	0.10	−0.12	−0.06	0.09	0.08
Predictability	PR	0.65	−0.02	0.07	0.34	0.14	−0.10	0.44
Cognitive Stress	CS	−0.13	0.84	0.16	0.15	0.04	0.11	−0.05
Stress	ST	−0.16	0.83	0.19	−0.05	0.14	0.12	−0.02
Depressive Symptoms	DS	−0.16	0.82	0.12	0.16	0.02	0.14	−0.05
Somatic Stress	SO	−0.15	0.82	0.14	0.11	0.07	0.01	−0.06
Burnout	BO	−0.14	0.81	0.24	−0.12	0.15	0.11	−0.05
Sleeping Troubles	SL	−0.06	0.77	0.13	−0.12	0.08	0.09	−0.01
Self-Rated Health	GH	0.24	−0.42	−0.15	0.28	0.06	0.04	−0.20
Work Pace	WP	0.07	0.17	0.78	−0.21	0.02	0.01	0.04
Cognitive Demands	CD	0.20	0.11	0.74	−0.09	0.14	−0.01	0.14
Emotional Demands	ED	−0.07	0.29	0.73	0.16	0.18	0.13	−0.05
Demands for Hiding Emotions	HE	0.00	0.17	0.71	−0.10	0.10	0.17	0.14
Quantitative Demands	QD	−0.18	0.30	0.57	0.20	0.13	0.04	−0.26
Work–Life Conflict	WF	−0.13	0.37	0.48	0.05	0.28	0.23	−0.18
Control over Working Time	CT	0.05	0.02	−0.13	0.72	0.02	0.08	0.13
Variation of Work	VA	0.00	−0.01	−0.05	0.67	0.02	−0.10	−0.19
Role Conflicts	CO	0.01	0.16	0.21	0.10	0.84	0.07	0.10
Illegitimate Tasks	IT	−0.10	0.17	0.29	0.01	0.78	0.13	−0.06
Job Insecurity	JI	−0.10	0.20	0.09	0.07	0.05	0.82	0.01
Insecurity over Working Conditions	IW	−0.15	0.23	0.22	−0.12	0.16	0.78	−0.05
Possibilities for Development	PD	0.29	−0.11	0.21	0.17	−0.01	−0.03	0.79
Influence at Work	IN	0.19	0.10	0.43	0.55	0.12	0.03	0.78

Note: ^a^ Eigenvalue ≥ 1; total variance explained: 63.96%.

**Table 5 healthcare-13-00825-t005:** The results of convergent validity.

Factor	AVE	CR
Factor1	0.52	0.92
Factor2	0.59	0.89
Factor3	0.67	0.82
Factor4	0.71	0.85
Factor5	0.62	0.76
Factor6	0.61	0.76
Factor7	0.85	0.97

Note: AVE = average variance extracted; CR = composite reliability.

**Table 6 healthcare-13-00825-t006:** The result of discriminant validity: Pearson correlation and the square-root value of AVE.

	Factor1	Factor2	Factor3	Factor4	Factor5	Factor6	Factor7
Factor1	0.71						
Factor2	−0.24	0.77					
Factor3	−0.02	0.41	0.69				
Factor4	0.06	0.06	−0.13	0.53			
Factor5	−0.08	0.31	0.48	0.04	0.79		
Factor6	−0.23	0.36	0.37	−0.05	0.31	0.78	
Factor7	0.56	0.03	0.30	0.24	0.16	−0.04	0.60

Note: The diagonal data are AVE square-root values.

## Data Availability

All data generated or analyzed during this study are included in this published article and its Appendix A.

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
