# Peer review of "COPSOQ III in China: Preliminary Validation of an International Instrument to Measure Psychosocial Work Factors"

_healthcare, 2025, doi:10.3390/healthcare13070825_

Round 1

Reviewer 1 Report

Comments and Suggestions for Authors

The article presents a psychometric assessment of COPSOQ III in a Chinese sample, demonstrating a significant effort to validate the instrument. However, some areas need revision to strengthen the robustness and clarity of the study.

In the statistical analysis section you can introduce improvements such as: specifying the methods of extraction and rotation of the EFA; justify the choice of criteria for the elimination of items and the elimination of the 4 factors; present the detailed factor loadings.

It is suggested that it include information on convergent and discriminant validity; as well as deepening the discussion about the cultural and methodological differences that can influence the results.

It will also provide information about the experts who have evaluated the validity of the content, namely what their area of research is.

In the section where it presents the construct validity, it only mentions in the text four components and their respective items and I did not understand the designations of the other three missing components. Could there be better clarification?

In the text it refers (see Supplementary Table 2). I did not understand if there is any supplementary table or if it is a matter of giving the reader an indication to observe only what is in table 2. I suggest that the information about the table always comes before it and, in case you are referring only to the table and not to another supplementary one, indicate only (see Table 2). The same happens for other tables throughout the text.

In the discussion section there may also be room for improvement, namely, further deepening the implications of the results for research and practice; a greater connection of the results obtained in the study with the theory that supports COPSOQ III, as well as deepening the comparison with other validation studies of COPSOQ III.

The points I mention here only present points that I believe will improve the work they have done so far.

Thank you for your attention.

Author Response

Comments 1: The article presents a psychometric assessment of COPSOQ III in a Chinese sample, demonstrating a significant effort to validate the instrument. However, some areas need revision to strengthen the robustness and clarity of the study.

Response 1: Thank you very much indeed for all of your patience and kind comments on our manuscript. Accordingly, we have reorganized and modified some areas of the manuscript. Further revisions are included in the response to the subsequent comments.

Comments 2: In the statistical analysis section you can introduce improvements such as: specifying the methods of extraction and rotation of the EFA; justify the choice of criteria for the elimination of items and the elimination of the 4 factors; present the detailed factor loadings.

Response 2: Accepted and revised. We have added a note about how the exploratory factor analysis (EFA) was done: “EFA was performed using principal component analysis (PCA) for extraction and Varimax rotation with Kaiser normalization to achieve a clear and interpretable factor structure. ” The choice of criteria for the elimination is added as ”The criteria for item elimination were based on multiple considerations: (1) items with factor loadings below 0.40 were removed to ensure strong associations with the intended factors; (2) items with cross-loadings exceeding 0.30 on multiple factors were excluded to maintain discriminant validity; and (3) items that did not contribute meaningfully to the theoretical interpretation of the factors were discarded” in the Methods section.

Additionally, we have presented the detailed data of factor loadings in Table 4.

Comments 3: It is suggested that it include information on convergent and discriminant validity; as well as deepening the discussion about the cultural and methodological differences that can influence the results.

Response 3:  Accepted and revised. We have added the key information about the convergent and discriminant validity of the questionnaire.  Specifically, we have added a detailed introduction to convergent and discriminant validity in the Methods section.This change can be found on page 6 and lines 274~284 of the revised manuscript.

The revised manuscript now includes the results of the questionnaire's convergent and discriminant validity(3.5.3. Convergent and discriminant validity). Additionally, we have expanded the Discussion section, highlighting how these data might impact the findings and offering insights for future research.

Comments 4: It will also provide information about the experts who have evaluated the validity of the content, namely what their area of research is.

Response 4: Accepted and revised. We have added a detailed description of the disciplines and research areas of the seven experts, as shown in Table 3.

Comments 5: In the section where it presents the construct validity, it only mentions in the text four components and their respective items and I did not understand the designations of the other three missing components. Could there be better clarification?

Response 5: Accepted and revised. We have added the designations of the Component 5~7 as “Negative Tasks”, “Workplace Insecurity“ , and “Developmental Influence“ in the respectively. This change can be found on page 10 and lines 361~365 of the revised manuscript.

Comments 6: In the text it refers (see Supplementary Table 2). I did not understand if there is any supplementary table or if it is a matter of giving the reader an indication to observe only what is in table 2. I suggest that the information about the table always comes before it and, in case you are referring only to the table and not to another supplementary one, indicate only (see Table 2). The same happens for other tables throughout the text.

Response 6: Accepted and revised. We have adjusted the text to introduce table information before referencing it (e.g., "as shown in Supplementary Table 2"). We have also ensured consistency in table references throughout the manuscript. These changes improve clarity and readability.

Comments 7: In the discussion section there may also be room for improvement, namely, further deepening the implications of the results for research and practice; a greater connection of the results obtained in the study with the theory that supports COPSOQ III, as well as deepening the comparison with other validation studies of COPSOQ III.

The points I mention here only present points that I believe will improve the work they have done so far.

Response 7: Accepted and revised. We have provided a more detailed explanation for the elimination of certain items from the COPSOQ III-Chinese instrument, including the following points: cultural relevance, statistical considerations, and expert feedback. These changes make sure that the shortened version of COPSOQ III is both culturally appropriate and psychometrically sound enough to be used in China, which also means that the results have even more important implications for research and practice.This change can be found on pages 14 and lines 423~494 of the revised manuscript. In the discussion section, we also added a comparison with validation studies of COPSOQ III from Canada, Brazil, Sweden, and Australia.This change can be found on pages 15 and lines 516~521 of the revised manuscript.

Reviewer 2 Report

Comments and Suggestions for Authors

First, I would like to thank you for the opportunity to read this work. This study represents a significant contribution to psychosocial risk assessment. It provides evidence regarding the reliability and validity of the Chinese long-version of COPSOQ 25 III, confirming that this instrument is useful for assessing psychosocial work factors in China.

Below, I offer some comments and suggestions, hoping they will enhance the paper.

In general, to improve the readability of this work, please avoid using long sentences. For instance, the sentence extends over five lines on page 1, lines 38-42. There are also typos across the manuscript (line 137: “woekers”).

INTRODUCTION

This introduction could be enhanced if the authors offered a clearer flow of ideas. There are key topics that must be addressed: (1) Definition of Psychosocial risks; (2) Psychosocial risks and data on Occupational health in China; (3) The measurement of Psychosocial risks – first worldwide and then specifically in China. In addition, by reading the abstract, the readers know this study was conducted in a specific activity sector. It would be interesting to add a section related to the health sector to the introduction.

MATERIALS AND METHODS

This section is, in general, well-written and easy to follow.

RESULTS

This section is also, in general, well-written and easy to follow.

DISCUSSION

The authors could strengthen the justification for the eliminated items and deepen the explanations in the trimmed version of COPSOQ 25 III. In addition, the authors could deepen the comparison between the Chinese version and the versions of other countries.

Comments on the Quality of English Language

In general, the quality of the English language is fine, but proofreading is necessary to fix some minor issues. 

Author Response

Comments 1: First, I would like to thank you for the opportunity to read this work. This study represents a significant contribution to psychosocial risk assessment. It provides evidence regarding the reliability and validity of the Chinese long-version of COPSOQ 25 III, confirming that this instrument is useful for assessing psychosocial work factors in China. Below, I offer some comments and suggestions, hoping they will enhance the paper.

Response 1:  Thank you very much for your constructive comments concerning our manuscript. Further revisions are included in the response to the subsequent comments.

Comments 2: In general, to improve the readability of this work, please avoid using long sentences. For instance, the sentence extends over five lines on page 1, lines 38-42. There are also typos across the manuscript (line 137: “woekers”).

Response 2: Accepted and revised. We have modified ”woekers” into ” workers” and go over the same once again to emphasize this point. Besides, we have modified the long sentences in original lines 38-42 as ”In 1984, the International Labour Organization (ILO) defined psychosocial factors (hazards) as the interactions between and among work environments, job content, organizational conditions, and workers’ capacities, needs, cultures, and personal extra-job considerations. It could influence workers’ health, work performance, and job satisfaction through perceptions and experience in the workplace. ” Additionally, We have revised a lengthy sentence, which originally stated: “Exposure to psychosocial hazards, such as discrimination, inequality, overwhelming workload, and job insecurity, may result in serious physical and psychological complications if the body under-recovers, including higher rates of cardiovascular diseases[6], burnout[7], musculoskeletal diseases[8], depressive symptoms[9], dementia[10], sleeping troubles[11], post-traumatic stress disorder (PTSD)[12], and suicide[13].”It is now divided into shorter sentences: “Exposure to psychosocial hazards can lead to serious physical and psychological complications. These hazards include discrimination, inequality, overwhelming workload, and job insecurity. If the body does not recover adequately, the risks increase for conditions such as cardiovascular diseases, burnout, and musculoskeletal disorders[8]. Other potential effects include depressive symptoms[9], dementia[10], sleep disturbances[11], post-traumatic stress disorder (PTSD)[12], and even suicide[13].”

Comments 3: INTRODUCTION

This introduction could be enhanced if the authors offered a clearer flow of ideas. There are key topics that must be addressed: (1) Definition of Psychosocial risks; (2) Psychosocial risks and data on Occupational health in China; (3) The measurement of Psychosocial risks – first worldwide and then specifically in China. In addition, by reading the abstract, the readers know this study was conducted in a specific activity sector. It would be interesting to add a section related to the health sector to the introduction.

Response 3: Accepted and revised. We have modified the structure to make a clearer flow of ideas in the introduction section, which lists the key topics in the following order: (1) Definition of psychosocial risks—Paragraph 2; (2) Psychosocial risks and data on occupational health in China—Paragraph 3; (3) The measurement of psychosocial risks—first worldwide and then specifically in China—Paragraph 4. In addition, we have added a section (Paragraph 5) related to the health sector to the introduction in the revised manuscript.

Comments 4: MATERIALS AND METHODS

This section is, in general, well-written and easy to follow.

Response 4: Agree. Thanks for your comments.

Comments 5: RESULTS

This section is also, in general, well-written and easy to follow.

Response 5: Agree. Thanks for your comments.

Comments 6: DISCUSSION

The authors could strengthen the justification for the eliminated items and deepen the explanations in the trimmed version of COPSOQ 25 III. In addition, the authors could deepen the comparison between the Chinese version and the versions of other countries.

Response 6: Accepted and revised. In the pages 14 and lines 423~494 of revised manuscript, we have provided a more detailed explanation for the elimination of certain items from the COPSOQ III-Chinese instrument include the following points: (1)Cultural Relevance. Some items were removed because they were deemed culturally inappropriate or difficult to comprehend within the Chinese context. (2)Statistical Considerations. For example, the Cronbach’s Alpha if Item Deleted (CAID) was greater than the overall Cronbach’s Alpha of each item were removed to enhance the internal consistency of the scale.The exact dimensions were deleted as “Job Engagement”, “Horizontal Trust”, “Vertical Trust”, and “Self Rated Health” with factor loading <0.04 and multiple loads. (3)Expert Feedback. Certain items were identified as less relevant or redundant in the Chinese workplace context through expert panel reviews. These modifications ensure that the trimmed version of COPSOQ III is both culturally appropriate and psychometrically robust for use in China.

In the discussion section, we also added a comparison with validation studies of COPSOQ III from Canada, Brazil, Sweden, and Australia.

Comments 7: In general, the quality of the English language is fine, but proofreading is necessary to fix some minor issues. 

Response 7: Agree. We have carefully reviewed the entire manuscript to correct spelling mistakes and modify sentence structure.

  1. Response to Comments on the Quality of English Language

Point 1: The English could be improved to more clearly express the research.

Response 1: We appreciate your attention to the linguistic aspects of our manuscript. In response, we have carefully revised the language, paying attention to sentence structure, vocabulary choices, and tone. Furthermore, we invited a senior officer from the WHO to review the English expressions in this manuscript.

Reviewer 3 Report

Comments and Suggestions for Authors

An extremely valuable research piece which is addressing the hitherto ignored aspects of the worker and interaction with workplace environment.

If found appropriate by the authors, some background(very brief) may be given of the various models that form the basis of the instrument. Most researchers have used those in isolation. This could be a good addition in the knowledge regarding synthesising various existing models to create a more robust, holistic one.

Additionally , it would be interesting to see the differences in the results if the sample size of Males and Females was kept the same .Research has proven significant  differences in perception of males and females in various conditions of work and society.

Some reframing of the sentences may be needed(Line49).Some spelling errors were also noticed(workers as 'woekers'-line 137).It is suggested that the authors go over the same once again.

Comments on the Quality of English Language

Needs a thorough read to check out spelling errors and sentence construction.

Author Response

Comments 1: An extremely valuable research piece which is addressing the hitherto ignored aspects of the worker and interaction with workplace environment. If found appropriate by the authors, some background (very brief) may be given of the various models that form the basis of the instrument. Most researchers have used those in isolation. This could be a good addition in the knowledge regarding synthesising various existing models to create a more robust, holistic one.

Response 1: Accepted and revised. COPSOQ covers the most prominent work environment theories, including demand control, social support, effort rewards, job demands,resources, work-family conflict, social capital, socio-technical, etc. We have added a more detailed background introduction of the above models. This change can be found on pages 2~3 and lines 73~99 of the revised manuscript.

Comments 2: Additionally , it would be interesting to see the differences in the results if the sample size of Males and Females was kept the same. Research has proven significant differences in perception of males and females in various conditions of work and society.

Response 2: Accepted and revised. Sample sizes for males and females already exhibit differences in our study as 74.00% of the participants were female and 26.00% male. We have added that “The scores of male HWs were significantly higher than those of female HWs in the following dimensions: quantitative demands, emotional demands, influence at work, control of working time, role conflict, illegitimate tasks, work-life conflict, cognitive stress, and depressive symptoms (P<0.05). The scores of female HWs were statistically significantly higher than those of male HWs in the following dimensionss:  development possibility, meaning of work, role clarity, social support from colleagues, sense of community at work, commitment to the workplace, job satisfaction, and organizational justice (P<0.05). These results prove significant differences in perception of males and females in various conditions of work and society.” in lines 317-325 of the revised manuscript.

In response, further analysis would be conducted to explore deepening differences in perception of males and females in various conditions of work and society in the following article.

Comments 3: Some reframing of the sentences may be needed(Line49).Some spelling errors were also noticed(workers as 'woekers'-line 137).It is suggested that the authors go over the same once again.

Response 3: Accepted and revised. We have modified ”woekers” into” workers and go over the same once again to emphasize this point. This change can be found on page 4 and lines 162 of the revised manuscript.

Comments 4: Needs a thorough read to check out spelling errors and sentence construction.

Response 4: Agree. We have carefully reviewed the entire manuscript to correct spelling mistakes and modify sentence structure. For instance, we have modified the long sentences in original lines 38-42 as ”In 1984, the International Labour Organization (ILO) defined psychosocial factors (hazards) as the interactions between and among work environments, job content, organizational conditions, and workers’ capacities, needs, cultures, and personal extra-job considerations. It could influence workers’ health, work performance, and job satisfaction through perceptions and experience in the workplace”. Additionally, We have revised a lengthy sentence, which originally stated: “Exposure to psychosocial hazards, such as discrimination, inequality, overwhelming workload, and job insecurity, may result in serious physical and psychological complications if the body under-recovers, including higher rates of cardiovascular diseases[6], burnout[7], musculoskeletal diseases[8], depressive symptoms[9], dementia[10], sleeping troubles[11], post-traumatic stress disorder (PTSD)[12], and suicide[13].”It is now divided into shorter sentences: “Exposure to psychosocial hazards can lead to serious physical and psychological complications. These hazards include discrimination, inequality, overwhelming workload, and job insecurity. If the body does not recover adequately, the risks increase for conditions such as cardiovascular diseases, burnout, and musculoskeletal disorders[8]. Other potential effects include depressive symptoms[9], dementia[10], sleep disturbances[11], post-traumatic stress disorder (PTSD)[12], and even suicide[13].”

  1. Response to Comments on the Quality of English Language

Point 1: The English could be improved to more clearly express the research.

Response 1: Accepted and revised. We have carefully revised the language, paying attention to sentence structure, vocabulary choices, and tone. Furthermore, we invited a senior officer from the WHO to review the English expressions in this manuscript.

Round 2

Reviewer 1 Report

Comments and Suggestions for Authors

I appreciate the changes made. It seems to me that the work has undergone significant improvement inputs.